# Neural Agents Struggle to Take Turns in Bidirectional Emergent Communication

**Valentin Taillandier**
ENS PSL, Inria
valentin.taillandier@gmail.com

**Dieuwke Hupkes**
Meta AI Research
dieuwkehupkes@fb.com

**Benoît Sagot**
Inria
benoit.sagot@inria.fr

**Emmanuel Dupoux**
Meta AI Research
dpx@fb.com

**Paul Michel**[*]
ENS PSL, Inria
pmichel31415@gmail.com

## Abstract

The spontaneous exchange of turns is a central aspect of human communication. Although turn-taking conventions come to us naturally, artificial dialogue agents struggle to coordinate, and must rely on hard-coded rules to engage in interactive conversations with human interlocutors. In this paper, we investigate the conditions under which artificial agents may naturally develop turn-taking conventions in a simple language game. We describe a cooperative task where success is contingent on the exchange of information along a shared communication channel where talking over each other hinders communication. Despite these environmental constraints, neural-network based agents trained to solve this task with reinforcement learning do not systematically adopt turn-taking conventions. However, we find that agents that do agree on turn-taking protocols end up performing better. Moreover, agents that are forced to perform turn-taking can learn to solve the task more quickly. This suggests that turn-taking may help to generate conversations that are easier for speakers to interpret.

## 1 Introduction

Natural conversations involve a rapid exchange of utterances where speakers coordinate on-the-fly to avoid talking over each other. This *turn-taking* phenomenon is ubiquitous across cultures (Stivers et al., 2009) and is even found in some forms of animal communication (Pika et al., 2018; Demartsev et al., 2018). The ability to engage in spontaneous turn-taking develops early in humans, even before linguistic competence (Nguyen et al., 2021) and allows us to hold fluent conversations with very little downtime between utterances (Heldner & Edlund, 2010). In contrast, fluid turn-taking is difficult to replicate in artificial dialogue systems. Modern conversational agents often rely on explicit cues, for instance pressing "enter" in text-based chatbots, the use of specific wake-words (Gao et al., 2020), or long silences of pre-determined length (Skantze, 2021).

The goal of this paper is to provide a testbed for studying the conditions under which artificial agents may develop a turn-taking convention to resolve a cooperative task. We describe a simple two-player game where agents observe

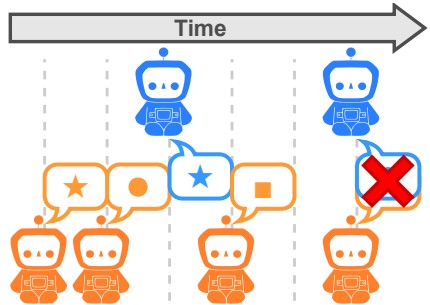

Figure 1: Illustration of our proposed game. Both agents can exchange utterances through a shared communication channel. At each step of the conversation, agents can decide to either speak or stay silent. However, information cannot be transmitted if both agents decide to speak at the same time.

partial views on an object which they must reconstruct. Agents can exchange information by emitting symbols across a shared communication channel over multiple rounds. The game exhibits two

---

[*]Now at DeepMind.

key features: first, at each round agents can decide to either talk or stay silent. However, if both agents decide to talk at the same time, they are not able to hear the other agent's message. Second, agents get a higher score if they solve the task in fewer rounds. This creates an explicit pressure towards a protocol that allows the agents to solve the task quickly while minimizing overlap.

In experiments, we find that simple neural-network-based agents trained with reinforcement learning do not consistently develop natural turn-taking strategies. However, agents that do develop a turn-taking protocol are able to achieve a much higher score, sometimes solving the task perfectly. To shed light on this finding, we perform an in-depth analysis of an asymmetric version of the game, where one agent has all the information. We show that in this case, there is an optimal strategy that does not rely on turn-taking. However, we demonstrate empirically that agents fail to solve the game when they are forced to use this strategy. In contrast, agents that are forced to use strategies in multiple turns rapidly achieve almost perfect accuracy.

## 2 Related Work

Schegloff (2000) attributes the first description of turn-taking to Goffman (1955), although the term "turn-taking" itself was coined much later (Yngve, 1970). Since then, and following early seminal work in the 70s (Duncan, 1972; Sacks et al., 1978), turn-taking has been the subject of considerable study under the umbrella of "conversation analysis" (Levinson, 1983). For example, researchers have sought to characterize linguistic and paralinguistic cues involved in organizing turns (Stephens & Beattie, 1986; Kendon, 1967; Clancy et al., 1996), to understand the time-scales involved in turn changes (Stivers et al., 2009; Heldner & Edlund, 2010) or to identify turn-taking conventions in non-human primates (Rossano, 2013; Rossano & Liebal, 2014; Demartsev et al., 2018).

Research in automated dialogue systems dates back to early efforts in the 60s and 70s (Weizenbaum, 1966; Bobrow et al., 1977). Particularly relevant to our work is a line of research on task-based methods, which formulates dialogue as a reinforcement learning problem to satisfy some user-relevant task (Walker, 2000; Levin et al., 2000). After the success of early deep-learning based models (Vinyals & Le, 2015; Li et al., 2016), most state-of-the-art systems (Adiwardana et al., 2020; Zhang et al., 2020) are now generally built on top of large pretrained models (Radford et al.; Lewis et al., 2020). Research in dialogue systems focuses primarily on text-based models, and spoken dialogue systems are generally implemented as speech recognition/generation scaffolding around a text-based core (Chen et al., 2018). However, there have been recent on addressing phenomena specific to spontaneous spoken dialogue such as filled pauses (Székely et al., 2019) or non-verbal vocalizations and turn changes (Nguyen et al., 2021).

Our work belongs to a long line of research devoted to the use of computational models for simulating the emergence of language (Batali, 1998; Cangelosi & Parisi, 2002). With the advent of deep learning, there has been a surge of interest in emergent communication among neural-network agents trained with reinforcement learning (Lazaridou et al., 2017; Foerster et al., 2016; Lazaridou & Baroni, 2020). The overwhelming majority of research in this area focuses on unidirectional sender-receiver communication (see Havrylov & Titov (2017); Bouchacourt & Baroni (2018); Chaabouni et al. (2020) *inter alia*) based on variants of the Lewis signaling game (Lewis, 1969; Skyrms, 2010). Although several authors have studied bidirectional communication (Kottur et al., 2017; Graesser et al., 2019), alternating turns are usually hard-coded into the game.

## 3 Method: A Testbed for the Emergence of Turn-Taking

There is evidence in the conversational analysis literature that turn-taking systems in human languages almost universally tend to favor (1) few instances of overlapping speech and (2) minimal pauses between turns (Stivers et al., 2009). In this section, we describe a communication game based on a variant of the Lewis signaling game (Lewis, 1969) which encapsulates these features.

### 3.1 Game Description

The game is played between two agents, Agent 1 and Agent 2. Before every episode of the game, an object $x$ is drawn according to distribution $p$ from a pool $\mathcal{X}$ and both agents observe partial views on the object, $\hat{x}_1$ and $\hat{x}_2$. The goal of the game is for the agents to communicate enough information to reconstruct the original input $x$. Throughout the paper, we will experiment with two versions of

the game: a **symmetric** variant where both agents observe partial views, and an **asymmetric** variant where one agent (by convention, Agent 2) has access to the full object ($\hat{x}_2 = x$).

### 3.1.1 CONVERSATION OVER A SHARED CHANNEL

During an episode, conversation between both agents proceeds over $T$ communication steps. At every step $t \in \{1, \ldots, T\}$, each agent $i \in \{1, 2\}$ must choose one of two actions:

- SILENCE: the agent says nothing
- SPEAK: the agent emits a symbol $s_t^i$ taken from a finite vocabulary $V = \{w_1, \ldots, w_{|V|}\}$. The agent chooses which symbol to send (there are therefore $|V|$ possible SPEAK actions).

Agents choose their respective actions synchronously and independently: they cannot coordinate to choose their action based on the other agent's action at the same time-step.

To emulate the presence of a single auditory channel, we introduce an additional environmental constraint depending on the actions chosen by the agents. Two situations may occur:

1. if at least one agent stays SILENT, communication succeeds. In this case, both agents are able to observe each other's actions, including any symbol sent.
2. both agents may attempt to SPEAK at the same time. We call this situation **overlap**: in this case, we do not allow the agents to perceive each other's actions directly, to simulate failure to transmit information. We detail specific strategies for modeling overlap in Section 3.1.2.

After each step of the conversation, agents can make a guess as to the original object $x$, based on the communication history they have observed and receive a reward $r_t$ for this communication step. Importantly, agents are trained to maximize their *total reward* $R = \sum_{t=1}^{T} r_t$. This means that agents are encouraged to reconstruct the correct object (and thus get maximal reward) as soon as possible, thus creating a pressure towards efficient communication (Rita et al., 2020).

### 3.1.2 MODELING OVERLAPS

Central to our game is the special handling of overlaps: agents cannot simply communicate information simultaneously. We implement this constraint by perturbing the symbols emitted by each agent. We experiment with several ways to corrupt the emitted symbols to represent a failure of communication that may result from agents speaking over each other:

- **Noise**: instead of observing the symbol emitted by their communication partner, each agent observes a special <noise> symbol. Both agents are aware that they have attempted to communicate but they do not observe the exact message sent by each other at this specific time-step.
- **Misunderstanding**: another possibility is that agents misunderstand each other in case of overlap. We implement this by substituting the emitted symbols with random symbols from the vocabulary.
- **Walkie-Talkie**: in this more restrictive setting, the communication channel behaves akin to a walkie-talkie device: when agents speak simultaneously, they perceive the other's action as SILENT. This is a more punitive case of overlap because the agents are not able to detect that they are overlapping.

In most experiments, we adopt the **Noise** model. However, we will show that the choice of the overlap model can have large consequences on the emergent protocol.

### 3.1.3 OBJECTS AND PARTIAL VIEWS

To facilitate analysis and enable fine-grained control of the difficulty of the game, in the remaining of this paper we consider games where the underlying objects $x$ are simple attribute-value vectors (Kottur et al., 2017; Chaabouni et al., 2020).[1] Specifically, each object $x$ is a vector of $N_a$ attributes, each of which can take $N_v$ distinct values, for a total of $N_v^{N_a}$ possible combinations. We represent

---

[1]Although the game could be performed on other objects, such as partially obstructed images (Graesser et al., 2019).

a partial view $\hat{x}$ on an object $x$ by masking $N_m$ of its attributes. At every episode of the game, we sample an object $x$ and randomly mask some of its attributes according to a distribution $\delta_m(\cdot|x)$ to obtain partial views $\hat{x}$.

In the asymmetric version of the game, only one partial view is sampled ($\hat{x}_1$) since Agent 2 observes the original object. In the symmetric variant, both partial views $\hat{x}_1$ and $\hat{x}_2$ are obtained by sampling masks independently for both.

## 3.2 MODELING THE AGENTS

### 3.2.1 GENERAL ARCHITECTURE

During an episode of the game, agents must achieve two tasks: they must exchange information across the communication channel, and they must be able to reconstruct the original object $x$ based on their conversation history. For this purpose, each agent $i$ must implement two functions:

- $\pi^a(a_t \mid S^i_{t-1})$, an *action policy* which defines a distribution over one of the $|V| + 1$ possible actions (either stay silent or emit one of the $|V|$ symbols)
- $\pi^r(x'_t \mid S^i_{t-1})$, a *reconstruction function* which defines a distribution over possible reconstructions of the original input $x$

Both distributions are conditioned on $S^i_{t-1}$, the state of the conversation up until step $t-1$ from the point of view of agent $i$. $S^i_{t-1}$ encapsulates (1) the original partial view $\hat{x}_i$ observed by the agent, (2) the history of all past actions of the agent $a^i_{<t}$ and finally (3) the symbols received from the other agent $j$, $\hat{s}^j_{<t}$. Recall that the latter may not correspond exactly to the symbols *emitted* by the other agent $j$ if the agents overlapped.

In practice we implement these functions with a shared conversation encoder $h(S^i_{t-1})$ which maps the conversation state $S^i_{t-1} := (a_{<t}, \hat{s}^j_{<t}, \hat{x}_i)$ onto a continuous representation (for instance by means of a recurrent neural network). Both the action policy and the reconstruction function are functions of this encoding (see Figure 2). The action policy is parameterized as a class-based softmax (Goodman, 2001), where we first predict the type of action, either SILENT or SPEAK. If the latter is chosen, a separate softmax operator predicts the emitted symbol. We factor the reconstruction function into $N_a$ independent classifiers, one for predicting the value of each attribute (out of $N_v$ candidates).

### 3.2.2 STANDARD TRAINING OBJECTIVES

During training, each agent is then trained to maximize two objectives, a conversation objective $J_{\text{conv}}$ with respect to its action policy and a reconstruction objective $J_{\text{rec}}$ with respect to its reconstruction function:

$$J_{\text{conv},i} = \mathbb{E}_{\substack{x \sim p \\ \hat{x}_1, \hat{x}_2 \sim \delta_m}} \mathbb{E}_{\mathbf{S} \sim \substack{\pi^a_1(\cdot|\hat{x}_1) \\ \pi^a_2(\cdot|\hat{x}_2)}} \left[ R^1(\mathbf{S}) + R^2(\mathbf{S}) \right]$$

$$J_{\text{rec},i} = \mathbb{E}_{\substack{x \sim p \\ \hat{x}_1, \hat{x}_2 \sim \delta_m}} \mathbb{E}_{\mathbf{S} \sim \substack{\pi^a_1(\cdot|\hat{x}_1) \\ \pi^a_2(\cdot|\hat{x}_2)}} \sum_{t=1}^{T} \log \pi^r_i(x \mid S^i_t)$$

where $\mathbb{E}_{\mathbf{S} \sim \pi^a_{\theta_1}(\cdot|\hat{x}_1), \pi^a_{\theta_2}(\cdot|\hat{x}_2)}$ denotes the expectation taken over all conversations sampled from the agents' policies conditioned on $\hat{x}_1$ and $\hat{x}_2$. Both agents learn to maximize the joint reward $R^1 + R^2$, making the game fully cooperative. We will refer to this objective as the **Standard** objective.

To encourage agents to communicate about their missing information, we experiment with an alternative objective where agents must not only reconstruct their masked attributes, but also predict which attributes are masked for their interlocutor. We call this objective **Reciprocal**, since agent $i$ must be able to reconstruct $\hat{x}_j$ and not $x$.

## 4 EXPERIMENTAL SETTINGS

### 4.1 GAME PARAMETERS

In all experiments, inputs are vectors of $N_a = 10$ attributes with $N_v = 16$ possible values, resulting in $\approx 10^{12}$ total combinations. Unless specified otherwise, agents are optimized on a training set of $10^6$ examples. We also sample a validation and test set of 1,000 examples each. We set the number of masks $N_m = 2$. During training, the positions of the masked attributes are sampled uniformly on-the-fly. At evaluation time, we exhaust all possible combinations of masks for the first agent. In the symmetric setting, we sample another pair of mask independently for the second agent.

We set the size of the vocabulary to $|V| = 8$. Importantly, we are careful to choose a communication channel that is small enough to dis-

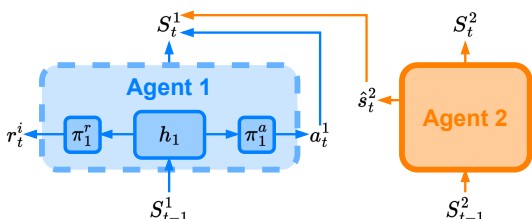

Figure 2: Overview of a step of the game from the point of view of Agent 1. The agent takes in $S_{t-1}^1$ and tries to reconstruct the object, yielding reward $r_t^1$. Simultaneously, the agent samples an action $a_t^1$ and observes symbol $\hat{s}_t^2$ emitted by agent 2. The conversation state $S_t^1$ is updated based on both agent's actions.

qualify trivial strategies where each agent simply describes the entirety of its partial view. Indeed, a single agent needs $N_v \log_{[|V|+1]} N_a$ steps to fully describe a vector of $N_a$ attributes with $N_v$ values using a vocabulary of size $|V|$.[2] With a vocabulary size of 8 this corresponds to $\approx 13$ steps per agent (26 in the symmetric setting). Therefore we limit the size of the channel to $T = 8$ in the asymmetric variant and 16 in the symmetric variant. This means that to solve the task, agents must leverage the fact that they share information about the problem if they are to solve the task successfully.

## 4.2 EVALUATION METRICS

We evaluate agents by their ability to reconstruct the original object at the end of the conversation. We report the average **accuracy** over all masked attributes. Note that in the symmetric version of the game, it is possible that some attributes are masked for both agents, meaning that the original value cannot be reconstructed. We exclude these occurrences from the computation, which means that the maximum accuracy is always 100%.

We also analyze the resulting conversations in terms of number of turns and overlaps, following standard conversation analysis taxonomy (Sacks et al., 1978). Specifically, a **turn** starts when an agent $i$ is speaking and the other agent $j$ isn't. The turn lasts up until agent $i$ becomes silent. If the next agent speaking is still agent $i$, the turn continues and any silence is counted as an "intra-turn pause". However, if the next agent speaking is agent $j$, the turn ends before the first silence, and agent $j$'s turn starts whenever it starts speaking. If there are intermediate steps between the two turns where both agent are silent, they are counted as a "gap" in the conversation. Therefore, each timestep between the start of the first turn and the end of the last turn can be counted either as agent $i$'s turn, agent $j$'s turn, or a gap. Additionally, we also record all timesteps where agents speak over each other (**overlap**).

## 4.3 AGENT IMPLEMENTATIONS

We implement the conversation encoder $h_i$ of each agent as single layer LSTM networks (Hochreiter & Schmidhuber, 1997) with hidden dimension 512 and input dimension 256. At each step $t$, agent $i$ encodes the state of the conversation as $h_t^i = \text{LSTM}(h_{t-1}^i, e_{a_{t-1}^i} + e_{\hat{s}_{t-1}^j})$ where $h_{t-1}^i$ is the hidden state corresponding to the previous timestep and $e_{a_{t-1}^i}$ and $e_{\hat{s}_{t-1}^i}$ are embeddings of the previous action $a_{t-1}^i$ and the symbol $\hat{s}_{t-1}^i$ received from the other agent at step $t-1$.

At the beginning of the conversation, the values of the partial views on the inputs, $\hat{x}_i$ are encoded as $N_a$ 256-dimensional embeddings, one for each attribute. For each attribute we learn $N_v + 1$ 256-dimensional embeddings, one for each values and one to represent the masked attribute. The concatenated $N_a \times 256$ dimensional representation of $\hat{x}_i$ is then fed into a fully connected layer mapping to a 512-dimensional encoding. This input encoding is used to initialize the hidden state of the recurrent network.

---

[2]The basis of the logarithm is $|V| + 1$ because agents can transmit information not only by emitting one of the $|V|$ symbols, but also by staying silent.

|        | Accuracy | Overlaps | #Turns |        | Accuracy | Overlaps | #Turns |
|--------|----------|----------|--------|--------|----------|----------|--------|
| Stand. | $92.63_{\pm2.55}$ | $0.00_{\pm0.00}$ | $1.00_{\pm0.00}$ | Stand. | $50.60_{\pm6.24}$ | $1.42_{\pm0.56}$ | $7.00_{\pm1.29}$ |
| Recip. | $90.10_{\pm12.17}$ | $0.77_{\pm0.72}$ | $2.05_{\pm0.23}$ | Recip. | $83.63_{\pm6.91}$ | $3.07_{\pm0.89}$ | $5.34_{\pm1.45}$ |

(a) Metrics for the **asymmetric** game.      (b) Metrics for the **symmetric** game.

Table 1: Metrics for both **asymmetric** and **symmetric** game with both standard and reciprocal objective. We report standard deviation across 20 runs.

## 4.4 AGENT TRAINING

Agents are trained to maximize their objectives using gradient descent with minibatching. At each step of the optimization process, we sample objects $x$ uniformly from the training set, and partial views $\hat{x}_1$ and $\hat{x}_2$ according to the masking distribution $\delta_m(\cdot \mid x)$. We then unroll the conversation by repeatedly sampling from each agent's policy until the maximum number of steps have been reached, yielding conversation $\mathbf{S} = \left(S_t^1, S_t^2\right)_{t=1\ldots T}$.

We can obtain gradient estimates for $J_{\text{rec}}$ by automatic differentiation. For maximizing $J_{\text{conv}}$ with respect to the policies $\pi^a$, we must resort to reinforcement learning. Specifically, we estimate $\nabla_{\theta_i} J_{\text{conv}}$ using REINFORCE (Williams, 1992), a policy gradient algorithm. We set the rewards to be the reconstruction log probability of both agents $R_{\mathbf{S}}^i = \sum_{i=1}^{T} \log \pi_i^r \left(x \mid S_t^i\right)$. We use the average reward in a minibatch as a baseline to reduce the variance of the REINFORCE updates (Sutton et al., 1999), and add an entropy maximization term to encourage exploration (Williams & Peng, 1991). Our implementation is written in Pytorch (Paszke et al., 2019), and is based on the EGG framework (Kharitonov et al., 2021).

## 4.5 TRAINING HYPERPARAMETERS

Agents are trained using the Adam optimizer (Kingma & Ba, 2015) with a learning rate of $0.001$, a batch size of $2048$ and a weight of $0.001$ for the entropy term. We train for a total of $600,000$ steps, and keep the pair of agents with the highest accuracy on the validation set. Unless indicated otherwise, all experiments are run for 20 different random seeds and we report both the mean and the standard deviation of each metric across all runs. See Appendix 7 for additional reproducibility.

## 5 RESULTS

### 5.1 ASYMMETRIC GAME: AGENTS DEFAULT TO UNIDIRECTIONAL COMMUNICATION

We first train agents to perform the asymmetric version of the game where Agent 2 has all the information. In Table 1a we report reconstruction accuracy (of Agent 1), the number of overlaps and the number of turns when agents are trained with both the **standard** and **reciprocal** objective. We find that agents trained with the standard objective adopt unidirectional communication strategies where only the agent that has all the information communicates (only one turn, no overlap). However, agents are not able to solve the game completely, consistently converging to about $92\%$ accuracy.

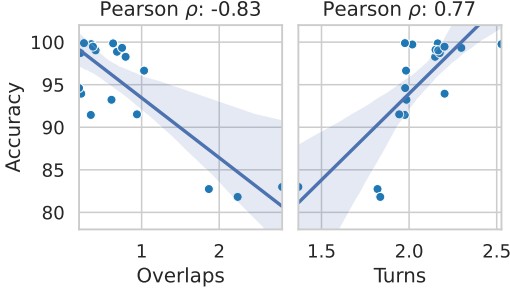

Figure 3: Correlation between turn-taking metrics and accuracy in conversations developed by agents for the asymmetric game ($p$-value $< 10^4$).

On the other hand, agents trained with the reciprocal objective are encouraged to communicate both ways to exchange information. We observe that agents generally develop protocols in more than one turn, albeit with some amount of overlap. Although agents reach lower accuracies on average, the variant across runs is very high. In fact, we find that more than half of the runs see agents achieve an accuracy greater than $98\%$ (by comparison, in the standard mode the best accuracy across all runs is $97.7\%$). To understand the source of this variability, in Figure 3 we report the accuracies reached by different agent pairs as a function of the number of turns and overlaps of their conversations. Conversations involving turn-taking (two or more turns, minimal overlap) correlate

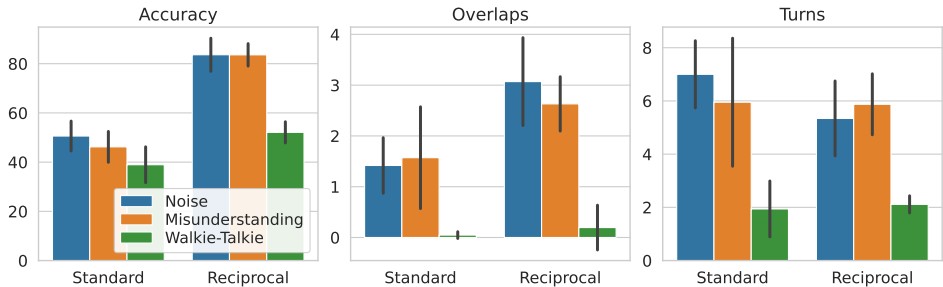

Figure 4: Influence of the overlap model on the emergent protocols for the symmetric game.

strongly with high accuracy. However, there are several outliers where agents are not able to develop efficient turn-taking and accuracy is much lower. In other words, while agents do not systematically agree on turn-taking protocols, those that do consistently perform almost perfectly at the game.

Appendix C showcases a successful "question-answer" based strategy, where Agent 1 first declares the masked positions and Agent 2 responds with the corresponding values.

### 5.2 SYMMETRIC GAME: NOT ALL TURN-BASED STRATEGIES ARE EQUAL

In the symmetric version of the game both agents observe masked inputs. Therefore it is not possible for the agents to rely solely on unidirectional communication. In Table 1b, we observe that agents achieve much lower accuracy than in the asymmetric game. In particular with standard training objective agents reach below $50\%$ accuracy. Interestingly, agents develop protocols involving many short turns ($\approx 7$ on average) with few overlaps.

Training with the reciprocal objective results in much higher accuracy. Agents still engage in turn taking, but the conversation involves fewer, longer turns ($\approx 5$ turns). We postulate that in this case agents adopt a similar strategy as in the asymmetric setting, first declaring their masked position and then announcing the missing values (see Appendix E for a qualitative example).

Overall, these results suggest that successful turn-taking does not necessarily entail success at the task (as for agents trained with the standard objective). Winning strategies rely on question-answers style conversations (agents trained with the Reciprocal objective).

### 5.3 INFLUENCE OF OVERLAP MODELING CHOICES

Up to this point, all experiments were performed with the "Noise" implementation of overlaps. However, it stands to reason that different models of overlaps may lead to different strategies. Figure 4 reports results for the symmetric game with all three implementations (see Appendix D for the asymmetric game). Although there is some variance, both Noise and Misunderstanding result in approximately the same behavior. On the other hand, agents trained in the Walkie-Talkie mode avoid overlaps at all costs: in this mode overlapping is much more punitive since agents that talk over each other are not aware that they overlapped. Consequently we find that agents struggle to perform conversations with multiple turns, and accuracy is much lower.

## 6 ANALYSIS

In this section we investigate why some agents adopting turn-taking conventions achieve higher accuracy. For the sake of simplicity, we focus our analysis on the simpler, asymmetric setting where one agent (namely Agent 2) has all the information.

### 6.1 OPTIMAL STRATEGIES

We describe two theoretical strategies which agents may adopt to solve the game while avoiding overlaps. For each, we can compute the length of conversation $L$ needed for the agents to fully recover the original object (see Appendix A for derivations).

- **Question-answers (QA)**: Agent 1 first communicates the position of here masks to Agent 2 (the "question"). Based on this information, Agent 2 (who knows the full object $x$) answers with the

values of the corresponding attributes. This strategy requires two turns, and so agents must agree on a turn-taking convention. The total length of the conversation is $L_{\mathrm{QA}} = \log \binom{N_a}{N_m} + N_m \log N_v$, where the first term corresponds to the length of the question and the second to the length of the answer. As shown in Section 5.1, we find that agents sometimes learn to adopt this strategy when they are trained with the reciprocal objective.

- **Error-correcting code (ECC)**: A more efficient approach is based on Reed-Solomon error-correcting codes (Reed & Solomon, 1960). In this strategy, both agents can agree on $N_m$ well-chosen linear encodings of the objects such that it is sufficient for Agent 2 to send the evaluation of these linear functions on $x$. Agent 1 can then provably reconstruct $x$ by evaluating the same linear functions on $\hat{x}$ and comparing the result with the message from Agent 2 (see Appendix B for a detailed explanation). The length of the conversation is then only $L_{\mathrm{ECC}} = N_m \log P_{N_a, N_v}$, where $P_{N_a, N_v}$ is the smallest prime number greater than $\max(N_a, N_v)$.

In our setting, the corresponding lengths (rounded to the next integer) are $L_{\mathrm{QA}} = 5$ and $L_{\mathrm{ECC}} = 3$. On the surface, ECC is much more efficient than QA: it requires 2 fewer steps to accomplish, and it only requires unidirectional communication.

## 6.2 Turn-taking Facilitates Learning

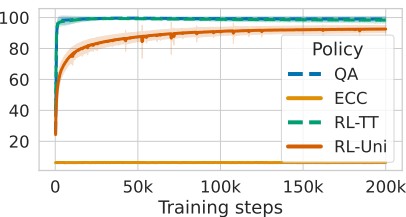

Figure 5: Validation accuracy throughout training with fixed policies. Conversations involving turn-taking (dashed lines) are learned much faster.

We have established that in the asymmetric setting, turn-taking is not the optimal strategy for solving the game as early as possible given the environmental constraints of a shared communication channel. This seems to contradict our results in Section 5.1, where we found that agents engaging in turn-taking performed better than agents relying on unidirectional communication. To understand this discrepancy, we compare the performance of the agents using any of these communication strategies. We compare four protocols: the two hard-coded strategies **QA** and **ECC**, and two strategies learned by agents through reinforcement learning. We select one policy where agents engage in turn-taking (**RL-TT**) and one where only unidirectional communication occurs (**RL-Uni**).

Figure 5 depicts the evolution of the reconstruction accuracy for pairs of agents trained purely to solve the task using a fixed policy. We find that turn-taking strategies not only enable the agent to reach the highest accuracy, but convergence is dramatically faster. In contrast, agents forced to adopt **RL-Uni** learn much slower. Finally, agents are not able to learn to solve the game using **ECC**, even though it is theoretically the most efficient strategy.

## 6.3 Discussion

These findings seem to discredit purely functional explanations for turn-taking as the most efficient communication strategy through a shared channel. Indeed, our theoretical analysis asserts that turn-taking is *not* optimal under these constraints. Nevertheless, according to Figure 5, conversations involving turn-taking are easier to interpret for agents than more sophisticated alternatives like ECC. Moreover, agents learn to decode these conversations much earlier. This begs the question whether turn-taking may have developed as a response to extraneous pressures beyond the efficient transfer of information. More future work is warranted to determine whether these findings generalize beyond our simple setting.

## 7 Additional Reproducibility Details

All experiments are run on single Nvidia V100 GPUs (16 or 32GB VRAM). Training takes about 4 hours (resp. 6 hours) for 100k steps for the asymmetric (resp. symmetric) experiments. A pair of agents totals 8,792k parameters and can run on a single GPU with a batch size of 2048.

Training parameters (learning rate and entropy coefficient) were chosen via a preliminary grid search in the asymmetric setting.

The code used for the analysis in this paper is publicly available on a GitHub repository.[3]

## 8 LIMITATIONS

Regarding agent modeling, we resort to simple LSTMs mostly for reasons of computational efficiency. However, neural architectures (*e.g.* Transformers; Vaswani et al., 2017) may imbue agents with different inductive biases and affect the resulting strategy. On a similar note, we only use a very simple reinforcement learning algorithm (REINFORCE with a baseline and entropy penalty). There are more sophisticated methods dedicated to multi-agent reinforcement learning in the literature (Papoudakis et al., 2021) such as counterfactual policy gradient (Foerster et al., 2018) or multi-agent variants of actor-critic algorithms (Yu et al., 2021). The use of such approaches may for instance enable agents to discover elaborate turn-based strategies (involving multiple questions and answers) without the use of an explicit auxiliary reciprocal guessing objective.

Modeling overlaps is the central feature of our proposed game. For the sake of simplicity, we only considered "discrete" implementations of overlap, replacing one symbol with another. A more elaborate alternative would be to represent symbols as actual speech signals (*e.g.* phonemes) and overlaps as the superposition of two signals. There is also some skepticism of the assumption that overlap significantly perturbs the original signal (see, *e.g.* Levinson, 2016). Moreover, sign languages also exhibit turn-taking, even though communication is not constrained by a shared auditory channel (Vos et al., 2016). This suggests that we could also model the effect of overlap as a difficulty (intrinsic to the agents) of producing and perceiving simultaneously (rather than a perturbation of the perceived signal), although it is not clear how this should be implemented. We leave the exploration of these alternatives to future work.

## 9 CONCLUSION

In this paper, we studied the emergence of turn-taking conventions in two-way communication. We proposed a simple language game where two agents must exchange information using a shared communication channel to solve a cooperative task. This enabled us to make several observations: first, neural network agents trained to play this game struggled to agree on effective turn-based strategies. Despite this fact, agents that did engage in turn-taking tended to achieve higher accuracy across the board. Detailed analysis showed that while turn-based communication is not necessarily optimal in theory, in practice it is much easier for agents to learn than more efficient alternatives. We hope that this work can foster future research on 1. understanding the origins of turn-taking in human communication and 2. developing conversational agents that can achieve turn-taking naturally.

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

# A    Strategies for Asymmetric Guessing

Let us focus on the asymmetric version of the game, where only one agent (hereafter the "Guesser") must reconstruct the object from a partial view $\hat{x}$, while the other agent (hereafter the "Oracle") has access to the full object $x$.

In this simplified scenario, we can describe theoretical conversation strategies that allow for solving the game without overlaps. We characterize each strategy by the effective amount of bits $I_{\texttt{guesser}}$ and $I_{\texttt{oracle}}$ of information that agents have to exchange after which the guesser can accurately reconstruct $x$.

**Naive strategy:**    A simple protocol is for the oracle to describe the entirety of the input $x$. This strategy requires little coordination: the guesser does not need to speak, she must simply parse the message to reconstruct $x$. Given that there are $N_v^{N_a}$ possible objects (distributed uniformly), classical information theory (Shannon, 1948) tells us that the number of bits needed to encode the objects is $I = -\log_2 \,^1/_{N_v^{N_a}} = N_a \log_2 N_v$.

$$I_{\texttt{guesser}} = 0$$
$$I_{\texttt{oracle}} = N_a \log_{|V|+1} N_v$$

**Question-answer:**    Naturally the naive strategy is sub-optimal: it does not leverage the fact that the guesser only needs to uncover the $N_m$ masked attribute. If the oracle knows which attributes the guesser is missing, she could simply send their values, which can be encoded in $N_m \log_2 N_v$ bits of information.

However, the guesser must first communicate the position of her $N_m$ missing attributes. The number of possible combinations of masks is given by the binomial coefficient $\binom{N_a}{N_m} = \,^{N_a!}/_{N_m!(N_a-N_m)!}$, and masks are chosen uniformly at random. Consequently the guesser's message needs to be $\log_2 \binom{N_a}{N_m}$ bits long. This is particularly efficient when $N_m$ is small in comparison to $N_a$. However, this strategy requires bidirectional communication: both agents must learn to take turns, and the oracle must learn to act conditioned on the guesser's message.

$$I_{\texttt{guesser}} = \log_{|V|+1} \binom{N_a}{N_m}$$
$$I_{\texttt{oracle}} = N_m \log_{|V|+1} N_v$$

**Error Correcting Code (ECC):**    Although the turn-taking strategy outlined above appears as a natural solution, there exists an even more efficient solution which doesn't necessitate any turn-taking whatsoever. Consider the simplest case where there is only $N_m = 1$ masked attribute, and assume that both agents agree on a numerical encoding of each value (the integers from 1 to $N_v$). In this case it is sufficient for the oracle to send the sum of the encoding of all values in $x$, modulo $N_v$. Upon reception, the guesser can simply compare this value with the sum of all $N_a - 1$ attributes it observes in $\hat{x}$ to retrieve the missing value. In this case the oracle only needs to send $\log_2 N_v$ bits of information for the guesser to be able to reconstruct $\hat{x}$.

This strategy can be extended to an arbitrary number of masks, borrowing ideas from the literature on error correcting codes. In particular, there is an encoding scheme based on Reed-Solomon codes (Reed & Solomon, 1960) in which the oracle evaluates well-chosen linear functions modulo $P_{N_a,N_v}$, the smallest prime number greater than both $N_a$ and $N_v$. Upon reception, the guesser can provably retrieve the values of the missing attributes by solving a system of $N_m$ linear equations in a finite field with $P_{N_a,N_v}$ elements. We refer to Appendix B for a detailed explanation of this code.

ECC is the most efficient strategy so far: it only necessitates communicating $N_m$ numbers of value up to $P_{N_a,N_v}$, leading to a total effective amount of bits of $I = N_m \log_2 P_{N_a,N_v}$. Moreover, it doesn't necessitate any turn taking at all: only the oracle needs to speak. However, the agent to agree upon a more sophisticated convention: in particular recovering the missing attributes from the message is not trivial.

$$I_{\texttt{guesser}} = 0$$
$$I_{\texttt{oracle}} = N_m \log_2 P_{N_a,N_v}$$

Moreover, the proposition 5.4 in (Dusart, 2018) shows that for any $x \geq 89693$, we can find a prime number $p$ so that $x < p \leq x(1 + {}^1/_{ln^3 x})$. This allows us to bound $P_{N_a, N_v}$ and shows that $I_{\texttt{oracle}} = N_m \log_2 P_{N_a, N_v}$ is never really far from $N_m \log_2 \max(N_a, N_v)$.

**Computing the length $L$ of a communication:** Knowing that agents are using discrete tokens in a vocabulary of size $|V| + 1$ (including silence), we can compute the effective length $L$ of a conversation with

$$L = \left\lceil \frac{I_{\texttt{guesser}}}{\log(|V| + 1)} \right\rceil + \left\lceil \frac{I_{\texttt{oracle}}}{\log(|V| + 1)} \right\rceil$$

This analysis can inform our choice of game parameters: for instance we choose $|V|$ and $T$ such that the naive strategy does not allow the agents to solve the game.

# B Error Correcting Code (ECC) Strategy

We describe in more details on the construction of the ECC protocol. Let's assume that $\hat{x}$ is the partial view of $x$, including a number of $N_m$ masks at the positions $k_1, \cdots k_{N_m}$.

The idea for the oracle agent is to compute $N_m$ values, $v_1, \cdots v_{N_m}$ defined as

$$v_k(x) := \sum_i x_i i^{k-1} \pmod{P_{N_a, N_v}} \tag{1}$$

where $P_{N_a, N_v}$ is the smallest prime number so that $P \geq \max(N_a, N_v)$. All the values are computed modulo $P_{N_a, N_v}$, which means that we only keep the remainder of the division by $P_{N_a, N_v}$. The amount of bits of information to transmit the values is therefore $I_{\texttt{oracle}} = N_m \log_2 P_{N_a, N_v}$.

Let us show that the $N_m$ values $v_1(x), \ldots v_{N_m}(x)$ are sufficient to reconstruct $x$ knowing $\hat{x}$. Let $v_1(\hat{x}), \ldots v_{N_m}(\hat{x})$ be the values computed by Agent 1 by applying Equation 1 to $\hat{x}$, counting masks as 0. By comparing $v_k(x)$ and $v_k(\hat{x})$, we obtain the following linear system

$$\begin{pmatrix} v_1(x) - v_1(\hat{x}) \\ \vdots \\ v_{N_m}(x) - v_{N_m}(\hat{x}) \end{pmatrix} = W \begin{pmatrix} x_{k_1} \\ \vdots \\ x_{k_{N_m}} \end{pmatrix} \tag{2}$$

where

$$W = \begin{pmatrix} 1 & \cdots & 1 \\ k_0 & \cdots & k_{N_m} \\ \vdots & \ddots & \vdots \\ k_0^{N_m} & \cdots & k_{N_m}^{N_m} \end{pmatrix} \tag{3}$$

is a matrix with elements in the field $\mathbb{Z}/P_{N_a, N_v}\mathbb{Z}$ of integers modulo $P_{N_a, N_v}$. $W$ is known as a Vandermonde matrix, and its determinant is given by

$$\det(W) = \prod_{1 \leq i < j \leq N_m} (k_j - k_i) \pmod{P_{N_a, N_v}}$$

Note that $0 < (k_j - k_i) < P_{N_a, N_v}$ for any $i, j$. This means that $\det(W)$ cannot be a multiple of $P_{N_a, N_v}$, since it is a prime number. Consequently, $\det(W) \neq 0$ modulo $P_{N_a, N_v}$, which means that $W$ is invertible in $\mathbb{Z}/P_{N_a, N_v}\mathbb{Z}$. To find the missing values, the guesser agent can simply solve the system by inverting $W$.

# C Example of a learned question-answer strategy

Figure 6 showcases a successful "question-answer" based strategy, where Agent 1 first declares the masked positions and Agent 2 responds with the corresponding values.

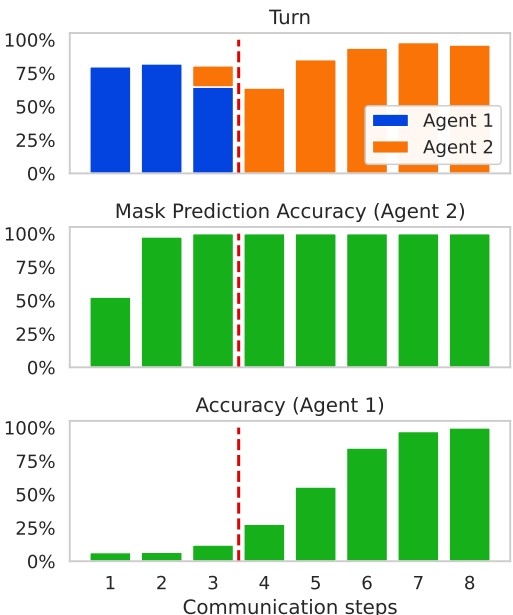

Figure 6: Example of question-answer strategy developed by a pair of agents in the asymmetric game with the reciprocal objective. During the first 3 steps, Agent 1 sends the position of the masked attributes: Agent 2 is able to correctly predict the masked positions. Afterward, Agent 2 responds with the missing values, and Agent 1's accuracy increases accordingly.

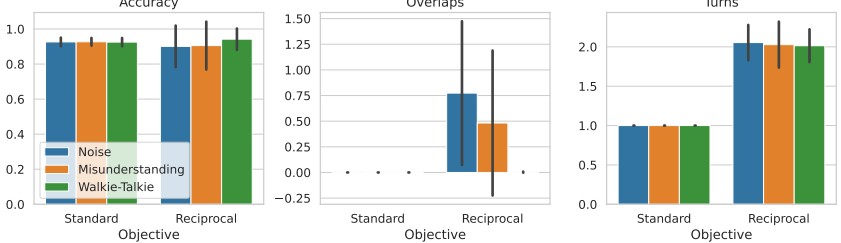

Figure 7: Influence of the overlap model on the emergent protocols for the asymmetric game.

## D    INFLUENCE OF OVERLAP MODEL IN THE ASYMMETRIC GAME

In Figure 7, we depict the influence of the choice of overlap model on the performance of the agents in the asymmetric game. Similarly to the symmetric game, Noise and Misunderstanding are relatively similar. Unsurprisingly, with the Walkie-Talkie model agents develop strategies without overlaps. However, contrary to the symmetric game here agents are able to develop a strategy in two turns even with this more punitive implementation of overlaps. We hypothesise that this is due to the smaller number of turns required to solve the game (only 2 versus at least 4 in the symmetric games). With only one turn change, it is easier to avoid overlaps.

## E    QUALITATIVE EXAMPLE OF QUESTION-ANSWER STRATEGY IN THE SYMMETRIC GAME

Figure 8 shows an example strategy developed by a pair of agents trained to play the symmetric game with the reciprocal objective. Agents follow a similar strategy as the asymmetric case (Figure 6 in the main text): in the first few steps, they exchange information about the positions of their masks. Based on this information, the rest of the conversation is devoted to communicating the values of these masked attributes.

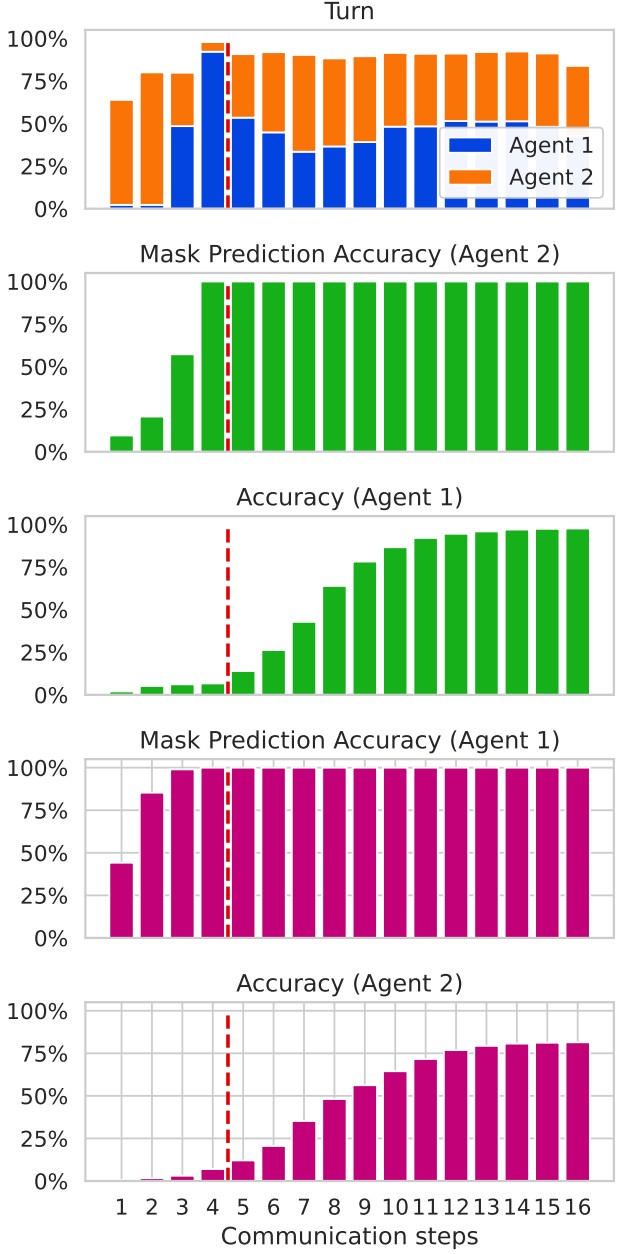

Figure 8: Example of question-answer strategy developed by a pair of agents in the symmetric game with the reciprocal objective. During the first 4 steps, Both agents communicate the positions of their masks. Afterwards, they each respond with the corresponding missing values.

