# OpenReview forum: "Neural Agents Struggle to Take Turns in Bidirectional Emergent Communication"
_ICLR.cc/2023/Conference — ICLR 2023 poster_

### Official Review · Reviewer_D8G5 · 2022-10-23

**Confidence:** 4
**Correctness:** 4
**Technical Novelty And Significance:** 3
**Empirical Novelty And Significance:** 4
**Recommendation:** 8

**Clarity, Quality, Novelty And Reproducibility:**

Clarity: The paper flows well and is well-motivated.

Quality: The experiments and analysis seem of high quality.

Novelty: The approach (game) is based on similar games for masked information discovery but its application in this setting to investigate turn-based strategies seems novel.

Reproducibility: Code is stated to be released upon deanonymization.

**Strength And Weaknesses:**

Strengths:
- The authors clearly establish the experimental setting and the possible methods of modeling overlap/interference are well justified and detailed (3.1.2).
- Interesting discussion on the difficulty of learning the optimal strategy in a variation of the conversational game and a demonstration that turn-taking is easier to adapt to

Weaknesses:
- A small comment: in the Misunderstanding setting, using garbled random symbols seems unnecessarily punitive, as misunderstandings seem often rooted in truth and at least snatches of what is meant to be communicated.
- It would be nice to see a comparison of how the studied effects may vary depending on vocabulary size (proxy for how complicated the language or modality of communication is) and model size

**Summary Of The Paper:**

The authors introduce a setting and game by which they study the emergence of conversational strategies when eliciting and discovering information. In particular, the authors show that commonly observed (among humans and animals) turn-taking behavior does not automatically arise in agents trained on the game, but that it does result in higher performance and that turn-taking is more easily adaptable and discoverable compared to other optimal strategies.

**Summary Of The Review:**

This is an interesting paper that investigates how turn-taking can evolve as a means of conversational strategy. The setting is clear and the authors investigate several paradigms of interference/penalties that map to realistic conversational situations. The results are interesting and contradict some explanations for the rise of turn-taking, and merit further study.

---

> ### Author Response · Authors · 2022-11-14
> **Discussion concerning the "Misunderstanding" setting and the size of vocabulary.**
>
> We thank the reviewer for their very encouraging comments.
>
>
>
> > in the Misunderstanding setting, using garbled random symbols seems unnecessarily punitive, as misunderstandings seem often rooted in truth and at least snatches of what is meant to be communicated.
>
>
>
> We agree with the reviewer that a more realistic model of this type of overlap the perturbed symbol would at least be similar in some way to the original symbol: for instance one might confuse the word “can” with “can’t” but not with “dog”. In our experiments we wanted to first focus on the most punitive instance first as they were more likely to elicit turn-taking behaviour. However, we do think that future work should investigate more realistic overlapping models grounded in the speech modality.
>
> > It would be nice to see a comparison of how the studied effects may vary depending on vocabulary size (proxy for how complicated the language or modality of communication is) and model size
>
>
>
> These are both interesting suggestions. Note that having access to a larger vocabulary with the same number of steps (and the same input space/number of masks) means that certain trivial strategies that do not require turn-taking may become available (see Appendix A for a discussion). We do think that it would be interesting to try simply a more complex setting with a larger vocabulary/number of steps (and a more complex space of inputs). However convergence of our agents was already slow and sometimes unstable even in our simpler setting. Therefore, additional work on the RL algorithm would be necessary to scale up to higher complexity.

---

### Official Review · Reviewer_31zg · 2022-10-24

**Confidence:** 4
**Correctness:** 3
**Technical Novelty And Significance:** 3
**Empirical Novelty And Significance:** 3
**Recommendation:** 6

**Clarity, Quality, Novelty And Reproducibility:**

The work targets an interesting aspect of communication and provides a really nice experiment to explore the problem, which has give scope for future work in the field. The experimental problem formulation is novel and have been explained properly in the paper. More implementation details would help in the reproducible aspect.

**Strength And Weaknesses:**

1. Created a simple and effective game to model the problem.

2. Showed experimental results for situations that both worked and did not work.

3. Could have used BERT instead of LSTM to generate embeddings.

**Summary Of The Paper:**

The authors have designed a simple language game to develop turn-taking conventions for neural agents. The game is a co-operative game, where two agents get a partial view of an object and the objective of the game is to reconstruct the original object with the lowest number of turns through mutual communication. Communication is done through a single channel that prevents both agents to speak at the same time, thus encouraging them to take turns while communicating. The partial view is constructed by masking some aspects of the original object. Overlaps were mostly modeled as noise, but other cases like misunderstanding and walkie-talkie were also experimented with.

The LSTM architecture was used to generate embeddings of the partial views of the agents. REINFORCE was used as the policy gradient algorithm for setting rewards. As for metrics, accuracy is used to measure if the prediction of the masked features. Experimental results show that agents trained under standard training objective tend to lean towards the agent with more information speaking most of the time and agents are able to solve the game 92% times, whereas with reciprocal objective, it is 98%.

**Summary Of The Review:**

1. It would be nice to see how the experiments perform if BERT is used to get embeddings of the agents. BERT is one of the state of the art models in NLP for generating embeddings.

2. Add more bottom padding to figure 3, as it hampers in reading.

---

> ### Author Response · Authors · 2022-11-14
> **Added more padding to figure 3 and response concerning BERT.**
>
> > It would be nice to see how the experiments perform if BERT is used to get embeddings of the agents. BERT is one of the state of the art models in NLP for generating embeddings.
>
>
>
> We thank the reviewer for this suggestion. It is unclear to us whether the reviewer is referring to the pretrained BERT model itself or the underlying transformer architecture.
>
>
>
> We did not consider BERT because it is a bidirectional model: each layer attends both to past and future token. As a result, it is not an appropriate model to implement our agent’s policies, which at any given step can only depend on previous actions and states. Moreover, we do not expect significant benefits from the fact that BERT has been pre-trained on natural language: the data in our synthetic game follows a very different distribution than natural language.
>
>
>
> However the reviewer may be referring to the transformer architecture, which is what BERT is based on. Indeed, the “decoder” version of the transformer could have been used as a policy instead of the LSTM. We chose to go with LSTMs for two reasons. First, they have been widely used in the emergent communication literature (virtually all works cited in the last paragraph in Section 2 use some variant of the LSTM). Second, in preliminary experiments we found that transformer-based agents were significantly more sensitive to hyper-parameter choices, in particular learning rate and entropy penalty coefficient. To keep the number of experiments and factors of variation in check, we decided to stick with LSTMs. However, we do think that subsequent work focusing on more complex tasks may benefit from revisiting this architectural choice.
>
> > Add more bottom padding to figure 3, as it hampers in reading.
>
>
>
> We thank the reviewer for pointing out this formatting issue. Upon re-reading the paper we found that padding seemed to be reasonable for Figure 3, but too short for Figure 4. Did the reviewer perhaps mean Figure 4?
>
>
>
> We have increased the padding below Figure 4 in our updated revision. However, please do let us know if there is still an issue with Figure 3 or any other figure.
>
> > More implementation details would help in the reproducible aspect.
>
>
>
> We attempted to give the salient implementation details in Section 4, in particular game parameters (4.1), model architecture (4.3), training algorithm (4.4) and hyper-parameters (4.5). Could the reviewer give us more specific examples of missing details which might make the paper easier to reproduce?
>
> Please also note that we commit to releasing our code (which is based on the opensource EGG and pytorch framework), which we hope will ease reproducibility.

---

### Official Review · Reviewer_qXss · 2022-10-25

**Confidence:** 3
**Clarity, Quality, Novelty And Reproducibility:** Detailed in the main review.
**Correctness:** 2
**Technical Novelty And Significance:** 2
**Empirical Novelty And Significance:** 2
**Recommendation:** 3

**Strength And Weaknesses:**

### Strengths:

- **************************Novelty:************************** The paper is unique in its selection of its problem to understand the emergence of turn-taking.
- ******************Clarity:****************** The paper, the methods, the metrics were quite easy to read and understand.
- The paper provides a good framework for how a communication channel might behave when overlapping messages are sent.

### Weaknesses:

- My main concern with the paper and the conclusions it draws are weak because of the simplistic task setting. For instance, it is unclear how the work might extend to future directions that the authors mention: 1. understanding the origins of turn-taking in human communication and 2. developing conversational agents that can achieve turn-taking naturally.
- The paper's conclusions (agents struggle to agree on turn-based strategies and that turn-taking agents achieve higher scores) seem pretty narrow. The paper did not convince me that they extend beyond the simple language game proposed by the authors.
- I don’t think the speculative claims in section 6.3 of functional vs cultural transmission are sufficiently supported by the analysis.

**Summary Of The Paper:**

The paper takes a look at emergent turn-taking in communication. To study this, the paper sets up a language game, which is a cooperative task, and information needs to be shared over a communication channel to be successful at the task. The authors introduce a condition that disrupts communication when turn-taking is violated. Through different task settings and reward engineering, the authors show how agents can be biased towards taking turns and how taking turns might improve the efficiency of learning a task.

**Summary Of The Review:**

The papers claims are not sufficiently supported by the tasks /agents tested. The tasks are too simplistic to improve our understanding of human communication or improving conversational agents.

---

> ### Author Response · Authors · 2022-11-14
> **Rephrased Section 6.3 and response to reviewer's concerns**
>
>
> We thank the reviewer for their comments. If our understanding is correct, the reviewer’s two key concerns about our work are that
>
> 1.  The setting is too simple
>
> 2.  The paper makes claims that are not well supported by the experiments due to the discrepancy between the simple setting and real human conversations (especially in Section 6.3)
>
>
>
>
> We respond to both points below
>
>
>
> > The setting is too simple
>
>
>
> As pointed out by the reviewer, the purpose of our work is to lay out and explore a novel problem area: the emergence of turn-taking convention in multi-agent communication. Accordingly, we made a conscious decision to start off with a very simple setting, which minimally implements 3 key conditions to the emergence of turn-taking: partial information, simultaneous communication, and overlap within a shared channel. Such a simplified setting allows us to answer interesting questions on the subject with minimal assumptions. This also makes it possible to carry out theoretical studies of possible communication protocols using tools from information theory (see Appendix A), which may not be feasible in more complex settings. Therefore, we believe that this work has value in and of itself, and are hopeful that it can serve as a basis for future work investigating more complex environments.
>
>
>
>
>
> Our work is part of a research theme that we may describe as the reverse engineering of language emergence and early language acquisition. Usually, hypotheses and claims on natural language motivated by cognitive sciences concerns are hard or impossible to capture “in vivo”. Oversimplified frameworks such as Lewis’ signaling game and its countless variations are intended to capture and exacerbate these hypotheses and claims. As the reviewer points out, these frameworks are the way to simple and unrealistic to let direct comparisons to humans, which would be purely speculative. Yet, we think interesting to observe synthetic agents converging in showing properties that we were expecting in humans.
>
>
>
> > The paper makes claims that are not well supported by the experiments due to the discrepancy between the simple setting and real human conversations (especially in Section 6.3)
>
>
>
> In the paper, we tried to be cautious with our language to avoid making overly general statements with regards to the relevance of our findings to natural language. We do not want to suggest that our results provide definitive novel insight on the emergence of turn-taking in human languages. However we do think that our results provide interesting insight: in particular, we were surprised to find that (1) agents did not develop turn-taking purely based off of the shared channel constraint and (2) there were more efficient strategies that did not rely on taking turns (see ECC, Section 6.1 and Appendix B) but were difficult for agents to learn.
>
>
>
> Following the reviewer’s suggestion, we rewrote Section 6.3 to tone down our comparison with human languages. We have removed the reference to iterated learning and cultural transmission, and rewrote the end of the paragraph to be more hypothetical:
>
>
>
> “Moreover, agents learn to decode these conversations much earlier. This begs the question whether turn-taking may have developed as a response to extraneous pressures beyond the efficient transfer of information. More future work is warranted to determine whether these findings generalize beyond our simple setting.”
>
>
>
> We hope that this addresses the reviewer’s concerns in this regard, however we are happy to discuss and reconsider other statements in the paper which the reviewer thinks might be problematic.

---

### Official Review · Reviewer_Nn7U · 2022-10-29

**Confidence:** 3
**Correctness:** 3
**Technical Novelty And Significance:** 2
**Empirical Novelty And Significance:** 2
**Recommendation:** 6

**Clarity, Quality, Novelty And Reproducibility:**

Exploring turn-taking for conversational agents through a language based game and exploring the different strategies to enable turn-taking aspects are novel in this work.

**Strength And Weaknesses:**

Strength and Weakness:
1. The basis of this work is interesting and studying turn taking in collaborative tasks in an understudied area. However, this work seems very preliminary and exploratory work and needs more analysis to be performed.
2. Why do the agents take lesser turns with partial information in the symmetric game? Shouldn't  higher accuracy indicate more communication/exchange of informations between the agents in Table 1(b)?


**Summary Of The Paper:**

In this work, the authors investigate the the ability of agents to engage in turn taking through a simple language based game. In this game, agents can emit information through a shared channel and using the information the agents needs to solve the task. The authors find that agents that develop turn taking achieve higher score

Contributions:
1. Exploration the turn taking strategies between agents through a language based game.
2. Initial results indicate agents that do turn taking are able to solve tasks quicker than non-collaborative agents.

**Summary Of The Review:**

In this work, the authors investigate the the ability of agents to engage in turn taking through a simple language based game.

---

> ### Author Response · Authors · 2022-11-14
> **We hope that our paper can serve as a stepping stone in this field.**
>
> > This work seems very preliminary and exploratory work and needs more analysis to be performed.
>
>
>
> As pointed out by the reviewer (and echoed by other reviewers), our work is very much exploratory in that it establishes the groundwork for investigating a new problem. We would like to highlight that our contributions include
>
>
>
> 1.  Proposing a minimal testbed including key features which precondition the emergence of turn-taking conversation (partial information, simultaneous communication and overlap) (Section 2)
>
> 2.  Defining associated metrics (Section 4)
>
> 3.  Perform experiments on several variations of our setting (symmetric/asymmetric, different overlaps) (Section 5)
>
> 4.  Provide empirical analysis of the emergent protocols (Section 6 and Appendix E)
>
> 5.  Perform theoretical analysis of possible communication protocols in this setting (Appendix A)
>
>
>
>
> We believe that these contributions are by themselves valuable and substantial, and hope that the paper can serve as a stepping stone for future research in the area.
> That being said, we are happy to discuss with the reviewer specific additional analysis which they believe should be performed to improve the paper.
>
>
> > Why do the agents take lesser turns with partial information in the symmetric game? Shouldn't higher accuracy indicate more communication/exchange of informations between the agents in Table 1(b)?
>
> If we understand the reviewer’s question correctly, they are asking why (in Table 1b) agents taking fewer turns (for example with the reciprocal objective) achieve higher accuracy than agents taking more turns (for example with the standard objective).
>
> Keep in mind that fewer turns does not necessarily mean less exchange of information given that individual turns may be longer. In fact, as shown in Section 6.1 agents can theoretically solve the game perfectly with about 4 turns using the “Question Answers” strategy. In fact, our qualitative analysis in appendices C and E suggests that agents trained with the reciprocal objective can adopt similar strategies (communicating first about the position of the masks, then about their content). In contrast, we find that agents trained with the standard objective tend to communicate in many short alternating turns.

---

### Author Response · Authors · 2022-11-14
**Rephrased Section 6.3  and Increased padding under Figure 4.**

We thank the reviewers for their feedback. We responded to their specific concerns in dedicated threads. Additionally, we have just uploaded a new revision of the paper. The two main changes in this version are:

- Rephrased Section 6.3 to tone down the comparison with the emergence of turn taking in natural language (Reviewer qXss)
- Increased padding under Figure 4 (Reviewer 31zg)

---

### Decision · Program_Chairs · 2023-01-20

**Decision:**

Accept: poster

**Justification For Why Not Higher Score:**

NA

**Justification For Why Not Lower Score:**

Good paper

**Metareview: Summary, Strengths And Weaknesses:**

This paper investigates the ability of AI agents to engage in turn taking through a language based game. This is an under-explored area and the paper's design of experiments and analysis is solid and inspire future work in this area. I am in favour of accepting the paper.

**Note From Pc:**

if the above contains the word "oral" or "spotlight" please see: "oral" presentation means -> notable-top-5% and "spotlight" means -> notable-top-25%. As stated in our emails, we are disassociating presentation type from AC recommendations